# Antimicrobial Effect of Natural Berry Juices on Common Oral Pathogenic Bacteria

**DOI:** 10.3390/antibiotics9090533

**Published:** 2020-08-24

**Authors:** Stefan Kranz, André Guellmar, Philipp Olschowsky, Silke Tonndorf-Martini, Markus Heyder, Wolfgang Pfister, Markus Reise, Bernd Sigusch

**Affiliations:** 1Department of Conservative Dentistry and Periodontology, Jena University Hospital, Friedrich-Schiller-University, An der Alten Post 4, 07743 Jena, Germany; Andre.Guellmar@med.uni-jena.de (A.G.); ph.ol@gmx.de (P.O.); Silke.Tonndorf-Martini@med.uni-jena.de (S.T.-M.); Markus.Heyder@med.uni-jena.de (M.H.); Markus.Reise@med.uni-jena.de (M.R.); Bernd.W.Sigusch@med.uni-jena.de (B.S.); 2Institute for Medical Microbiology, University Hospital Jena, Erlanger Allee 101, 07747 Jena, Germany; Wolfgang.Pfister@med.uni-jena.de

**Keywords:** ribes nigrum, ribes rubrum, rubus idaeus, vaccinium macrocarpon, chlorhexidine, antibacterial agents, cytotoxicity, oral plaque bacteria, periodontitis

## Abstract

(1) Background: Antimicrobial agents such as chlorhexidine (CHX) are commonly used in oral plaque control. However, sometimes those agents lack antimicrobial efficiency or cause undesired side effects. To identify alternative anti-infective agents, the present study investigated the antibacterial activity of all-fruit juices derived from blackcurrant, redcurrant, cranberry and raspberry on common oral pathogenic gram-positive and gram-negative bacteria (*Streptococcus mutans*, *Streptococcus gordonii*, *Streptococcus sobrinus*, *Actinomyces naeslundii*, *Fusobacterium nucleatum*, *Aggregatibacter actinomycetemcomitans*, *Porphyromonas gingivalis*, *Enterococcus faecalis*). (2) Methods: Antibacterial efficiency was evaluated by agar diffusion assay and in direct contact with bacteria in planktonic culture. Furthermore, cytotoxicity on human gingival fibroblasts was determined. (3) Results: Blackcurrant juice was most efficient at suppressing bacteria; followed by the activity of redcurrant and cranberry juice. Raspberry juice only suppressed *P. gingivalis* significantly. Only high-concentrated blackcurrant juice showed minimal cytotoxic effects which were significantly less compared to the action of CHX. (4) Conclusion: Extracts from natural berry juices might be used for safe and efficient suppression of oral pathogenic bacterial species.

## 1. Introduction

The health-promoting aspects of diets rich in natural berries have been discussed for quite some time. Most of these effects can be ascribed to the antioxidant activity of certain compounds that are usually found in natural berries in high concentrations [1,2,3,4]. The role of antioxidants, especially of berry phenols, in the prevention and treatment of various diseases has been underlined even by the WHO [5]. Aside from the berries’ phenols, other components such as vitamin C have strong antioxidant activity, too [6]. As already shown by our group, vitamin C also plays a distinct role in the prevention and treatment of various oral infectious diseases, such as periodontitis [7,8,9,10]. Recently our group showed that the additional supplementation of vitamin C significantly reduces the cytotoxic and apoptotic effects provoked by the oral pathogenic species *Porphyromonas gingivalis* on human gingival fibroblasts [8].

A sufficient intake of vitamin C is especially important for patients already suffering from periodontitis, who are commonly afflicted with reduced plasma vitamin C levels as well [9,10,11]. Because natural berries are essentially rich in vitamin C, the recommended daily dose could be met by merely consuming 80 g of blackcurrant, for instance [12].

Because of their high content in antioxidants, it was found that juices and extracts derived from natural berries also present strong anti-cancer activity. In this context, it was proven that juices prepared from blackcurrant, cranberry and raspberry were able to significantly decrease the proliferation rate of certain cancer cell lines [13,14].

Besides their anti-cancer activity, berry juices and extracts also have strong antimicrobial activity [15]. For example, it was shown that berry phenols are capable in selectively inhibiting the growth of different human pathogenic bacteria [16,17]. Additionally, it was proven that cranberry juices efficiently prevent oral plaque bacteria from co-aggregation and co-adhesion [18,19,20]. Recently it was found that proteins extracted from natural berry juices, in addition to their strong antimicrobial characteristics, also present high antifungal activity [21].

At present, chlorhexidine (CHX) in different concentrations and formulations is still recommended as the gold standard in chemical oral plaque control [22]. Application of this antimicrobial solution, however, is sometimes attended by some undesired side effects such as interference with taste, oral sensation of burning, dryness of the mucosa and discoloration of dental hard and soft tissues [23,24]. In addition, CHX has been shown to be cytotoxic, too [25].

Because of these reasons, natural plant extracts are constantly under investigation for evaluation of their antimicrobial capability [26,27,28]. Observations from these studies might help to design new and efficient anti-infective and bioactive agents with fewer side effects for safe and efficient microbial control.

So far, though, there is only minor information available that focuses on the antibacterial activity of extracts derived from natural berries on oral pathogenic bacteria. Therefore, the present in vitro study aimed on investigating the antibacterial effect of all-fruit juices obtained from blackcurrant (*Ribes nigrum* L.), redcurrant (*Ribes rubrum* L.), raspberry (*Rubus idaeus* L.) and cranberry (*Vaccinium macrocarpon Aiton*) on different gram-positive and gram-negative oral pathogenic bacteria. In addition, the cytotoxicity of these juices on human gingival fibroblasts (hGFs) was determined as well.

## 2. Results

### 2.1. Agar Diffusion Assay

It was found that juices derived from blackcurrant, red current and cranberry caused inhibition zones among all species tested. Blackcurrant juices induced the largest inhibition zones (Figure 1). Compared to treatment with CHX 0.2% (positive control), blackcurrant was even more efficient at inhibiting the species *P. gingivalis*. Furthermore, it was found that blackcurrant juice shows an inhibitory effect on the species *A. naeslundii*, *F. nucleatum* and *A. actinomycetemcomitans* that is comparable to the action of CHX.

Juices derived from cranberry showed efficient athibition activity, too. It was found that in case of *F. nucleatum* and *A. actinomycetemcomitans* cranberry juice induced inhibition zones that are comparable to the inhibitory effect of CHX, too.

The lowest inhibitory effect was observed for raspberry juice. However, still, raspberry juice was efficient at inducing inhibition zones among the species *S. gordonii*, *S. sobrinus*, *A. naeslundii* and *P. gingivalis*.

### 2.2. Direct Contact Test

The direct contact test was conducted in order to determine the antibacterial effect of the berry juices as a function of the incubation time and juice concentration. As already observed in the agar diffusion assay, blackcurrant and cranberry juice were of superior antibacterial activity.

As presented in Figure 2, direct contact with blackcurrant juice (concentration 90%/incubation time 60 s) caused total suppression of *S. gordonii*, *S. sobrinus*, *F. nucleatum*, *P. gingivalis* and *E. feacalis*. Compared to the positive control (CHX 0.2%) treatment with blackcurrant was even more efficient upon the species *S. gordonii*, *S. sobrinus* and *E. faecalis*. Additionally, direct contact of *S. sobrinus* with cranberry juice and incubation of the species *S. gordonii* and *E. faecalis* with redcurrant juice caused a stronger antibacterial activity compared to treatment with CHX.

The results of the present study show that the antibacterial behavior of the juices differed strongly among the species tested. While *S. sobrinus* and *S. gordonii* were significantly suppressed by juices of blackcurrant, redcurrant and cranberry (concentration 90%, incubation time 60 s), only minor reductive activity was observed for the species *A. naeslundii* and *S. mutans* (Figure 2).

Among all juices tested, raspberry juice showed the lowest antibacterial efficiency. Only *P. gingivalis* was significantly affected by this kind of juice.

Additionally, it should be recognized that among all agents tested, only treatment with blackcurrant juice (concentration 90%, incubation time 60 s) resulted in complete suppression of the species *E. faecalis* (Figure 2). Strong antibacterial effect on *E. faecalis* was also found for redcurrant juice. In contrast, treatment with 0.2% CHX resulted in bacterial suppression of only three log counts in total.

Furthermore, it was shown that the antibacterial effect was dependent upon the exposure time (Figure 3). It was found that an increase in incubation time resulted in stronger antibacterial activity. Among all species tested, *S. mutans*, *A. naeslundii* and *A. actinomycetemcomitans* were less affected by a prolonged incubation time. It was found that most of the bacterial species were already sufficiently suppressed after an exposure of only 20 s.

Moreover, it was observed that the antibacterial efficiency was also influenced by the juice concentration (Table 1). At concentrations of 50%, blackcurrant, redcurrant and cranberry juice still caused bacterial reduction at some species (≥10^3^ CFU/mL). Lower juice concentrations failed to induce any significant antibacterial effect when compared to the negative control (*p* < 0.5). In this study, a substance was rated antibacterial if there was a reduction in CFU/mL of at least three log counts.

The present study revealed that blackcurrant juice was most efficient at killing oral pathogenic bacteria (Figure 2). At concentrations of 90% and exposure times of 60 s, blackcurrant juice proved to suppress 5 out of 8 pathogenic species completely. Redcurrant and cranberry juice caused complete suppression of at least three species, which was also observed for the CHX control. The effect of the raspberry juice was restricted to *P. gingivalis* only.

### 2.3. Cytotoxicity

The cytotoxicity of all berry juices (90%), as well as of 0.2% CHX (positive control), was determined by MTT and NR test on human gingival fibroblasts (hGF) after exposure for 60 s (Figure 4). The results proved high loss of viability after exposure to CHX by either test, whereas no cytotoxicity was found for redcurrant, raspberry and cranberry juice. Blackcurrant juice caused an increase in cell activity in the MTT assay and revealed minor cytotoxicity in the NR test, which was statistically not significant.

In the present study, the cell morphology was analyzed by laser scanning microscopy after staining the actin filaments with phalloidine-rhodamine. The cells incubated with 0.2% CHX showed strong changes in morphology (Figure 5e). Evidently, those cells were arrested in the adherence phase (spherical) and were prevented from spreading further. Exposure to highly concentrated blackcurrant juice only caused minor changes in morphology, which is probably attributable to an accumulation of globular actin (Figure 5a). Cells treated with redcurrant, raspberry or cranberry juice at concentrations of 90% for 60 s did not result in an expression of morphological abnormalities (Figure 5b–d). Additionally, no morphological changes were observed among the cells incubated with DMEM (negative control) (Figure 5f).

## 3. Discussion

The aim of the present study was to determine the antibacterial effect of freshly prepared all-fruit juices of blackcurrant, redcurrant, raspberry and cranberry on common oral pathogenic species associated with the development of caries, periodontitis and endodontic infections. The antibacterial efficiency was tested by means of the agar diffusion assay and in direct contact to bacteria in planktonic solution.

As shown by the results, blackcurrant juice caused the strongest antibacterial effect of all juices tested. The strong antimicrobial effect of blackcurrant on pathogenic bacteria has been observed by other authors, too. For example, Puupponen-Pimia et al. showed that lyophilized powders of blackcurrant at concentrations of 2 mg/mL caused significant inhibition of *S. enterica* and *S. aureus* [17]. Furthermore, Miladinovic et al. published data showing that blackcurrant juice was sufficient in suppressing *L. monocytogenes*, *P. aeruginosa*, *S. aureus*, *B. cereus*, *S. enteritidis* and *E. coli* [4]. Ikuta et al. observed that extracts derived from blackcurrant reduce *S. pneumoniae* by 79% and *Haemophilus influenzae* type B by 99% [29].

In the present study, it was found that blackcurrant juice suppressed the gram-positive bacterial species *E. faecalis* completely (Figure 2). *E. faecalis*, in particular, often withstands treatment with customary disinfectants and is therefore frequently responsible for root canal treatment failures [30]. Compared to the action of blackcurrant juice, CHX revealed a rather low antibacterial effect on *E. faecalis* in the present study.

Besides blackcurrant, cranberry juice also proved to possess strong antibacterial activity. In particular, *S. sobrinus*, *F. nucleatum* and *P. gingivalis* were susceptible to treatment with cranberry juice. The significant antibacterial behavior of juices and extracts derived from cranberries has already been discussed by other authors, too. In this context, LaPlante et al. observed significant bacterial reduction among the gram-positive species *S. epidermidis*, *S. aureus*, methicillin-resistant *S. aureus* (MRSA) and *S. saprophyticus* after incubation with cranberry juice. However, gram-negative *E. coli* was unfortunately not merely affected in the referenced study [31]. Cesoniene et al. reported upon cranberry extracts being sufficient on different gram-positive and also gram-negative species, such as *E. coli* and *S. typhimurium* [32].

In addition, constituents of cranberries have also been investigated by various authors. Moreover, it was found that anthocyane and proanthocyanidine, especially, extracted from cranberries, are antibacterial, but that they fail to suppress the oral-pathogenic species *S. mutans*, and only partially inhibit *E. faecalis* [33]. Additionally, the non-dialyzable material (NDM) of cranberries, which is a sugar- and acid-free enriched complex of phenolic polymers (anthocyane, proanthocyanidine), was analyzed, but did not present any antibacterial activity on oral bacteria [34]. In contrast, other studies have shown that cranberry NDM is anti-adhesive and prevents different plaque bacteria from co-aggregation [18,19,35].

Furthermore, the present study revealed a strong antibacterial effect of raspberry juice on *S. sobrinus*, *A. naeslundii*, *P. gingivalis* and *E. faecalis* in the agar diffusion assay (Figure 1). This is supported by a study published by Cavanagh et al. that also reported on raspberry juice being antibacterial towards *E. faecalis* [36]. However, in direct contact, only *P. gingivalis* was significantly suppressed by raspberry juice in the present study (Figure 2). In conclusion, it can be assumed that raspberry juice is only of minor antibacterial activity on oral pathogenic species (Figure 4).

A great number of substances responsible for the antibacterial behavior of the berry juices have already been identified. In this regard, phenolics seem to be the most active components, occurring either as simple molecules, such as flavonoids, stilbenes and phenolic acids, or as complex phenolic polymers, e.g., tannins [37]. So far, it has been shown that especially flavonoids such as quercetin and myricetin, chlorogenic acid, caffeic acid, ferulic acid and various anthocyans and phenothiazines are of strong antibacterial activity, too [38,39,40,41].

Overall, there are already numerous components of natural origin that have been tested for their ability to disturb oral plaque formation. These include neovestitol and vestitol isolated from Brazilian red propolis, melanoidin and non-melanoidin components from coffee, magnolol and honokiol extracted from magnolia bark, organic acids such as succinic, malic, lactic, tartaric, citric and acetic acid from red and white wines, methyl chavicol, eugenol and methyl cinnamate from basil essential oils [42,43,44,45,46,47]. Additionally, it has been shown that the combination of antibiotics with ethanolic extracts from Salvadora persica and Cinnamomum zeylanicum results in a synergistic effect and causes a strong antimicrobial effect on different periodontal pathobionts [48].

One major finding of the present study can be seen in the efficiency of blackcurrant juice in killing *E. faecalis*, which even exceeds the action of CHX. Moreover, the blackcurrant juice was also of lower cytotoxicity compared to treatment with CHX. LSM evaluation revealed only minor changes in the cells’ morphologies after incubation with highly concentrated blackcurrant juice, which suggests an increase in accumulated globular actin. Furthermore, a rise in cellular activity was revealed in the MTT test, which is often attributed to stressing. The minor cytotoxic reactions of blackcurrant can probably be referred to the high number of polyphenols present in blackcurrants.

Among the limitations of the present study, an incubation time of only 60 s was chosen in order to observe the cytotoxicity. An increase in incubation times will probably allow more detailed interpretations. However, the present in vitro study focused on incubation times that were comparable to those commonly applied for CHX-containing products in everyday oral hygiene [22]. Detailed examinations with prolonged incubation times are needed and will be conducted in further observations.

In contrast to the tested berry juices, the present study revealed a rather strong cytotoxic effect for CHX after incubation for only 60 s. According to the MTT and NR assays, high loss of cell activity and a marked restriction in cell vitality were assessed for the hGFs after incubation with 0.2% CHX (Figure 5). Likewise, LSM evaluation showed heavy changes in the cells’ morphologies (Figure 5e). The cytotoxic effects of CHX on oral fibroblasts are already well known [25,49,50].

## 4. Materials and Methods

### 4.1. Preparation of Berry Juices

In the present study, the antibacterial and cytotoxic effects of different juices derived from the following berry species were observed: blackcurrant (*Ribes nigrum* L.), redcurrant (*Ribes rubrum* L.), raspberry (*Rubus idaeus* L.) and cranberry (*Vaccinium macrocarpon Aiton*). All fruits were obtained from controlled organic farming and harvested in 2013 in Germany.

The deep-frozen berries were defrosted within 30 min at 37 °C and blended for 1 min using a conventional food mixer (Tribest Personal Blender PB-250, Keimling Naturkost GmbH, Buxtehude, Germany). Subsequently, the received all-fruit juices were centrifuged (4800 rpm, 10 min) and the supernatants were collected and subjected to centrifugation for another 5 min at 13,200 rpm. The resulting juices were heat sterilized at 70 °C for 30 min. After sterilization, all juices were immediately deep frozen at −20 °C and stored in the fridge until use.

### 4.2. Bacterial Species

The study included the cariogenic species *Streptococcus mutans* ATCC 25,175-S.m., *Streptococcus gordonii* ATCC 10,558-S.g., *Streptococcus sobrinus* ATCC 33,478-S.s., *Actinomyces naeslundii* ATCC 19,039-A.n., the periodontopathogenic species *Fusobacterium nucleatum* ATCC 10,953-F.n., *Aggregatibacter actinomycetemcomitans* ATCC 33,384-A.a., *Porphyromonas gingivalis* ATCC 33,277-P.g., and the endopathogenic species *Enterococcus faecalis* ATCC 29,212-E.f.

*S. mutans*, *S. gordonii* and *S. sobrinus* were cultivated in tryptone soy bouillon (Carl Roth GmbH, Karlsruhe, Germany). *A. naeslundi*, *F. nucleatum*, *A. actinomycetemcomitans*, *P. gingivalis* and *E. faecalis* were cultivated in Schaedler bouillon (Oxoid LTD, Basingstoke, Hampshire, UK) supplemented with vitamin K (10 mg/L). All species were cultivated under anaerobic standard conditions (10% CO_2_, 10% H_2_, 80% N_2_) at 37 °C for 24 h. The bacteria were pelleted by centrifugation (3500 rpm, 5 min) and re-suspended in nutrient medium to an optical density (OD 546 nm) of 0.5.

### 4.3. Agar Diffusion Assay

For the agar diffusion assay, the bacterial suspensions (OD 0.5) were further diluted with PBS (Gibco, Life Technologies GmbH, Darmstadt, Germany) to a ratio of 1:100. Aliquots (100 μL) of *A. naeslundii*, *F. nucleatum*, *A. actinomycetemcomitans*, *P. gingivalis* and *E. faecalis* were plated onto Schaedler blood agar plates (6% defibrinated sheep blood, 0.1% Vit. K (10 mg/mL)), while those of *S. mutans*, *S. gordonii* and *S. sobrinus* were applied onto tryptone soy agar plates (Carl Roth GmbH, Karlsruhe, Germany). Two holes 9 mm in diameter were punched into each agar plate, and 100 µL of the respective juice pipetted into each hole. Physiological saline (0.9% NaCl) served as negative control, and 0.2% chlorhexidine solution (Meridol med, GABA GmbH, Lörrach, Germany) as positive control. The agar plates were afterwards cultivated under anaerobic standard conditions at 37 °C for 2 to 3 days, and those with P. gingivalis for 5 to 7 days. Subsequently, the respective inhibition zones were measured using a conventional ruler. For each bacterial species and juice, four plates with two punched holes each were prepared. Final juice concentrations of 90% were applied.

### 4.4. Direct Contact Test

To determine the antibacterial effect of the berry juices as a function of the exposure time (20 s, 40 s, 60 s), 10 μL of each bacterial suspension (0.5 OD 546 nm) was added to either 90 µL berry juice, 0.2% CHX (positive control) or nutrient solution (negative control). Final berry juice concentrations of 20%, 50% and 90% were maintained. After the respective exposure time, 100 µL of each sample was diluted in 900 μL PBS (in case of CHX, additionally supplemented with 0.07% lecithin solution) followed by centrifugation for 5 min at 4800 rpm, removal of the supernatant and re-suspension of the resulting pellet in 200 μL PBS. To determine the colony-forming units (CFU), decadal dilution series down to 10^−6^ were arranged. Each dilution was plated onto 6% Schaedler blood or tryptone soy agar, respectively. The agar plates were incubated under anaerobic standard conditions at 37 °C for 2 to 3 days, and those with *P. gingivalis* for 5 to 7 days. Six parallel samples per exposure time and juice concentration were run.

### 4.5. Cytotoxicity Test

For the cytotoxicity tests, human gingival fibroblasts (hGFs) were used (Ethics Committee Approval B1881-19/06). The cells were cultivated in DMEM (Dulbecco’s modified eagle medium) supplemented with 10% FBS (fetal bovine serum) and 1% AAS (antibiotic antimycotic solution) at 37 °C, with 5% CO_2_. The tests were administered with cells from the 6th to 8th passage. The cells were transferred to 96-well microtitration plates (10,000 cells/well) and cultivated for 24 h. Subsequently, the culture medium was removed, and the cells were exposed for 60 s to 100 µL of the respective juice at a final concentration of 90%. As positive control 0.2% CHX was applied. Eight parallel batches per test substance were processed. After incubation (60 s), the supernatants were removed and the cultures washed with 200 µL PBS each, overlaid with 100 µL culture medium and cultivated for another 24 h at 37 °C, 5% CO_2_. Cell viability was determined by applying the MTT-assay and NR-test as advice by the manufacturers.

In the present study, the morphology of the hGFs after incubation with the respective juices or controls was observed by laser scanning microscopy. Therefore, the hGFs were adhered to cover slips (approximately 8000 cells/cm^2^), transferred to 24-well microplates, and overlaid with 500 µL of DMEM each. After 2 h of incubation (adherence time), the cell culture medium was removed, and the cultures were incubated each with 300 µL of the respective juice for 60 s. As controls, 0.2% CHX (positive control) and DMEM (negative control) were used. Thereafter, the cultures were washed twice with 300 µL PBS each and again overlaid with 500 µL DMEM. After cultivation for 24 h at 37 °C, 5% CO_2_ the samples were fixated using PFA (paraformaldehyde 5%) and stained with phalloidine-rhodamine. Evaluation was performed using a LSM 510 Meta laser-scanning microscope (Carl Zeiss, Jena, Germany) equipped with a planapochromat 40/1.30 Oil Ph3 objective (Carl Zeiss). Rhodamine-labeled cells were excited with a diode laser (561 nm), and their emission passed a 565 nm long-pass filter set.

### 4.6. Statistical Analysis

The data was analyzed using the IBM SPSS Statistics program version 22 (SPSS, Chicago, IL, USA). Significant differences in the direct contact tests and the cytotoxic studies were checked by ANOVA (variance) analysis and subsequent post hoc test according to Dunnett. The significance level was *p* < 0.05. All samples were normally distributed, with common level of variance and drawn independently of each another.

## 5. Conclusions

The present study shows that freshly prepared all-fruit juices of blackcurrant, redcurrant and cranberry are antibacterial to most of the oral bacterial species tested (*S. gordonii*, *S. sobrinus*, *F. nucleatum*, *A. actinomycetemcomitans*, *P. gingivalis*, *E. faecalis*), with none or only very limited cytotoxic effects on human gingival fibroblasts. In future, extracts derived from natural berries might be used as substitutes in anti-infective agents for safe and efficient microbial control.

## Figures and Tables

**Figure 1 antibiotics-09-00533-f001:**
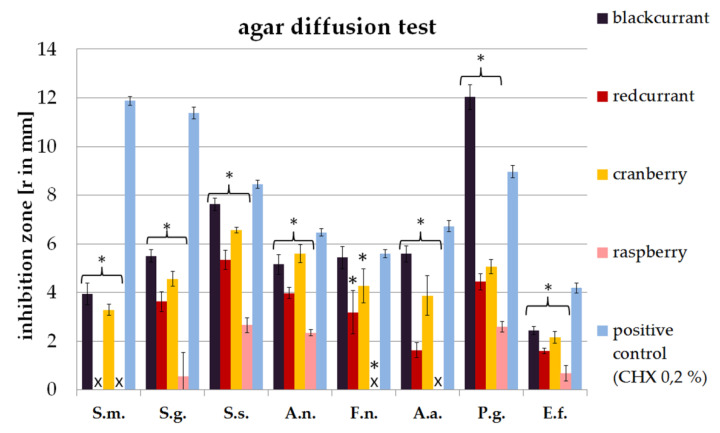
Antibacterial effect of the berry juices evaluated by an agar diffusion assay. Stars (*) indicate significant difference to the positive control (*p* < 0.05). X represents absence of any inhibition zones. (*Streptococcus mutans*-S.m.; *Streptococcus gordonii*-S.g.; *Streptococcus sobrinus*-S.s.; *Actinomyces naeslundii*-A.n.; *Fusobacterium nucleatum*-F.n.; *Aggregatibacter actinomycetemcomitans*-A.a.; *Porphyromonas gingivalis*-P.g.; *Enterococcus faecalis*-E.f.).

**Figure 2 antibiotics-09-00533-f002:**
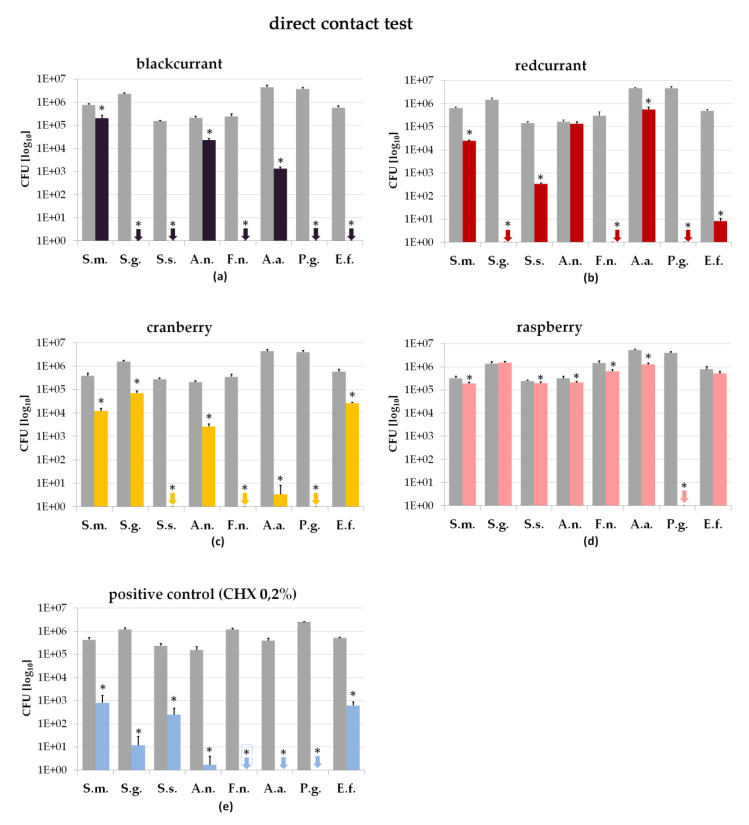
The results of the direct contact test are shown (juice concentration 90%/incubation time 60 s); gray column represents negative control (nutrition media); black column represents test agent in direct contact for 60 s; * asterisk indicates significant differences (*p* < 0.05) from the negative control. (**a**) Blackcurrant; (**b**) redcurrant; (**c**) granberry; (**d**) raspberry; (**e**) positive control (0.2% CHX) (*Streptococcus mutans*-S.m.; *Streptococcus gordonii*-S.g.; *Streptococcus sobrinus*-S.s.; *Actinomyces naeslundii*-A.n.; *Fusobacterium nucleatum*-F.n.; *Aggregatibacter actinomycetemcomitans*-A.a.; *Porphyromonas gingivalis*-P.g.; *Enterococcus faecalis*-E.f.).

**Figure 3 antibiotics-09-00533-f003:**
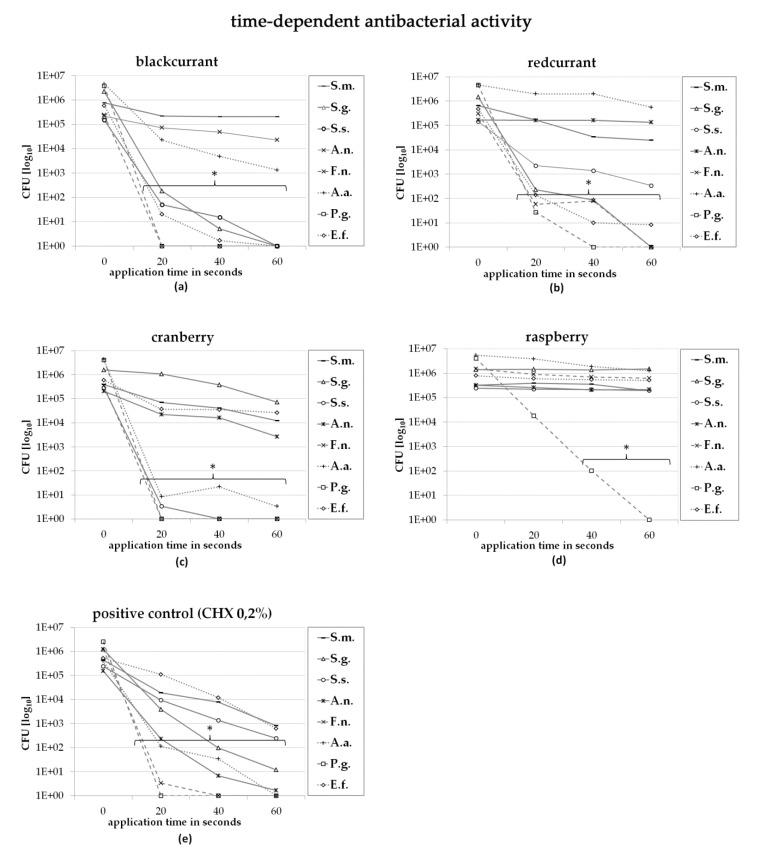
Direct contact test, time-dependence (incubation time 20 s, 40 s, 60 s; juice concentration 90%); * asterisked values show significant difference from the negative control. (*Streptococcus mutans*-S.m.; *Streptococcus gordonii*-S.g.; *Streptococcus sobrinus*-S.s.; *Actinomyces naeslundii*-A.n.; *Fusobacterium nucleatum*-F.n.; *Aggregatibacter actinomycetemcomitans*-A.a.; *Porphyromonas gingivalis*-P.g.; *Enterococcus faecalis*-E.f.).

**Figure 4 antibiotics-09-00533-f004:**
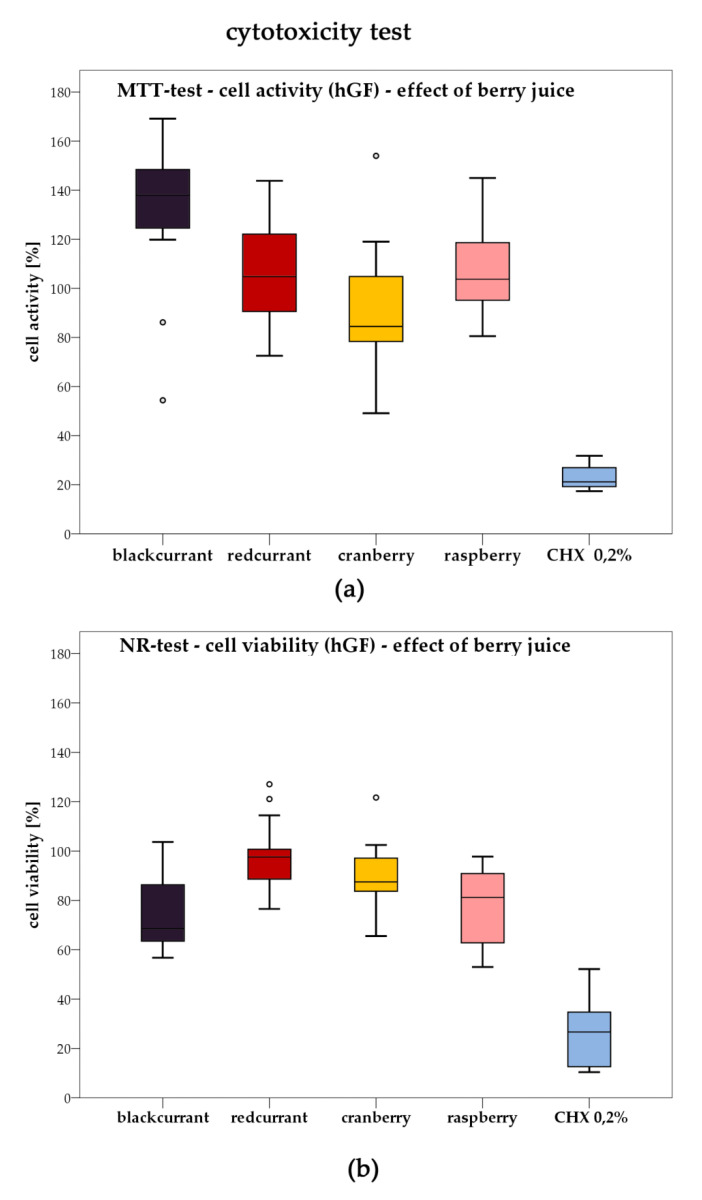
Cytotoxicity of highly concentrated berry juices (90%) after 60 s of incubation on human gingival fibroblasts (hGF); (**a**) cell viability determined by MTT assay, (**b**) cell viability determined by NR assay.

**Figure 5 antibiotics-09-00533-f005:**
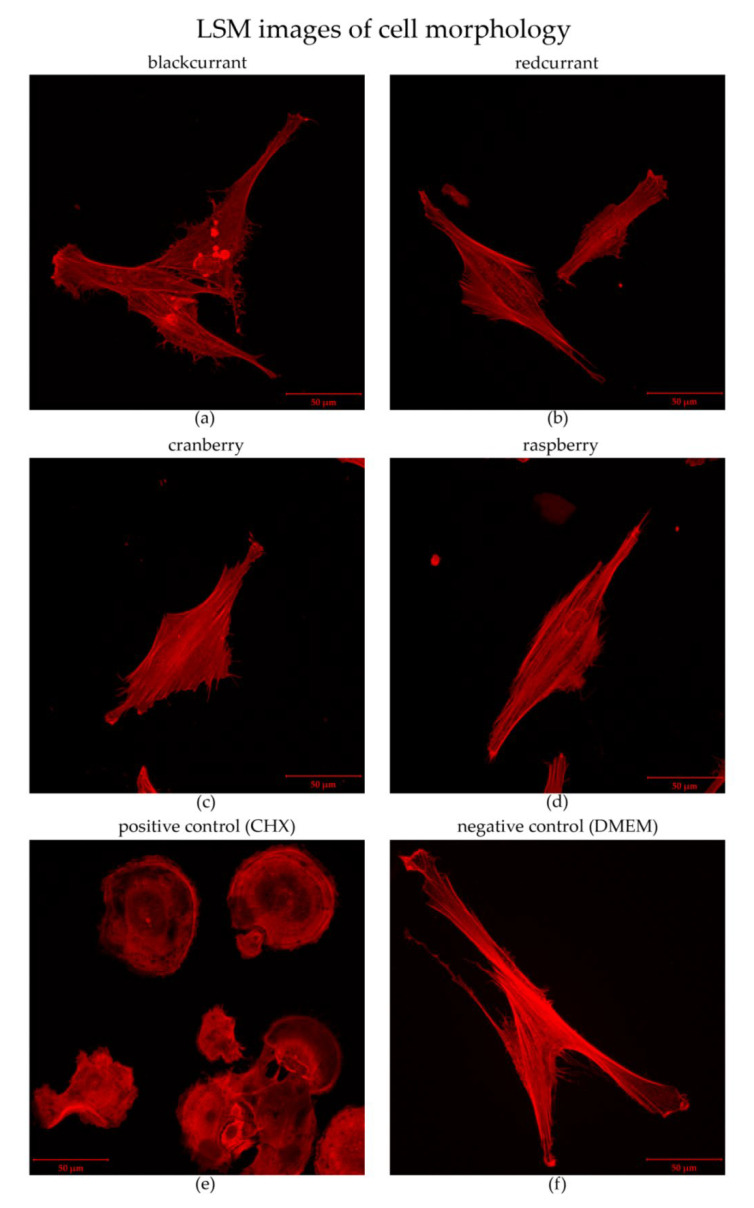
Morphology of human gingival fibroblasts (hGF) observed by laser scanning microscopy after direct contact with the respective test agent. Actin filaments were stained with phalloidine-rhodamine. (**a**) Blackcurrant; (**b**) redcurrant; (**c**) granberry; (**d**) raspberry; (**e**) positive control (0.2% CHX); (**f**) negative control (DMEM).

**Table 1 antibiotics-09-00533-t001:** Direct contact test—concentration-dependent antibacterial activity of berry juice (20%, 50%, 90%; exposure time 60 s) on various oral pathogenic bacterial species; no inhibitory effects (-); plate counts differ by > 10^3^ compared to untreated control (+); complete bacterial suppression (++). (*Streptococcus mutans*-S.m.; *Streptococcus gordonii*-S.g.; *Streptococcus sobrinus*-S.s.; *Actinomyces naeslundii*-A.n.; *Fusobacterium nucleatum*-F.n.; *Aggregatibacter actinomycetemcomitans*-A.a.; *Porphyromonas gingivalis*-P.g.; *Enterococcus faecalis*-E.f.).

Concentration-Dependent Antibacterial Activity								
Test Substance	S.m.	S.g.	S.s.	A.n.	F.n.	A.a.	P.g.	E.f.
**90% concentration**								
blackcurrant	**-**	**++**	**++**	**-**	**++**	**+**	**++**	**++**
redcurrant	**-**	**++**	**-**	**-**	**++**	**-**	**++**	**+**
cranberry	**-**	**-**	**++**	**-**	**++**	**+**	**++**	**-**
raspsberry	**-**	**-**	**-**	**-**	**-**	**-**	**++**	****-****
positive control (CHX 0.2%)	**-**	**+**	**-**	**+**	**++**	**++**	**++**	**-**
**50% concentration**								
blackcurrant	**-**	**-**	**-**	**-**	**+**	**-**	**++**	**+**
redcurrant	**-**	**-**	**-**	**-**	**-**	**-**	**-**	**+**
cranberry	**-**	**-**	**-**	**-**	**+**	**-**	**++**	**-**
raspsberry	**-**	**-**	**-**	**-**	**-**	**-**	**-**	**-**
positive control (CHX 0.2%)	**-**	**+**	**-**	**+**	**++**	**+**	**++**	**-**
**20% concentration**								
blackcurrant	**-**	**-**	**-**	**-**	**-**	**-**	**-**	**-**
redcurrant	**-**	**-**	**-**	**-**	**-**	**-**	**-**	**-**
cranberry	**-**	**-**	**-**	**-**	**-**	**-**	**-**	**-**
raspsberry	**-**	**-**	**-**	**-**	**-**	**-**	**-**	**-**
positive control (CHX 0.2%)	**-**	**+**	**-**	**-**	**+**	**+**	**++**	**-**

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
