# Peer review of "Antimicrobial Effect of Natural Berry Juices on Common Oral Pathogenic Bacteria"

_antibiotics, 2020, doi:10.3390/antibiotics9090533_

Round 1

Reviewer 1 Report

The paper is interesting and worth publishing, but we would suggest the authors to take into account the following critical observations:

Line 14: “Sometimes,  those  disinfections  are  afflicted  with  undesired  side  effects.” We would suggest to rephrase this sentence, particularly the words “disinfections” and “afflicted”.

Lines 236-237: the complete names of the species used here should be Ribes nigrum L., Ribes rubrum L., Rubus idaeus L., and Vaccinium macrocarpon Aiton.

Line 279: decadal dilution series down to 10-6 should have “-6” in superscript (10^-6).

Lines 71-83: various qualitative statements are made about the different juices for different target species but no p values are provided to support those statements.

Figures 1 and 2: whereas it is always preferable to report exact p values than mere p<0.05,  the conventional practice is to use one, two or three stars to indicate different degrees of statistical significance. The presence of only one star should be interpreted as in all cases p is lower than 0.05 but higher than 0.01 for instance? A three level system of indicating statistical significance, while still imperfect, would be better than the current one, using one star only.

Besides, the statistical methodology mentions ANOVA, but no information on checking ANOVA assumptions is made.

As shown by Figure 3, whereas some bacterial species were suppressed by a contact of only 60 seconds, for some a longer contact testing would have been relevant to understand whether they can be suppressed by longer contact times.

The use of only 60s for cytotoxicity assessment is much too short to allow any proper interpretation of the cytotoxicity findings. For many cytotoxicity tests hours or days are used. In this case one might speculate that to reflect the short time the juice will stay in the oral cavity a shorter time is needed, but even so, in our view it was preferable to have at least 5-10 minutes if not a few hours for a better understanding of the cytotoxicity. It is true that for such a short time (60s) the juices seem safer, but what if after 3 minutes there is no difference between them and CHX? This limitation should be discussed in the paper, as well as other limitations of the study.

I find the acknowledgment rather curious: the authors are thanking themselves for being authors.

Author Response

Dear Reviewer,

thank you for reviewing our manuscript and all the valuable comments that helped us to improve the scientific appearance of the manuscript. We tried to answer each comment to the best of our knowledge. Please find below a detailed list of our corrections:

Statement: The paper is interesting and worth publishing, but we would suggest the authors to take into account the following critical observations:

Line 14: “Sometimes, those disinfections are afflicted with undesired side effects.” We would suggest to rephrase this sentence, particularly the words “disinfections” and “afflicted”.

Answer: We rephrased this sentence. It is now: ”But, sometimes those agents lack in antimicrobial efficiency or cause undesired side effects.”

Statement: Lines 236-237: the complete names of the species used here should be Ribes nigrum L., Ribes rubrum L., Rubus idaeus L., and Vaccinium macrocarpon Aiton.

Answer: We corrected all species names as suggested.

Statement: “Line 279: decadal dilution series down to 10-6 should have “-6” in superscript (10^-6).

Answer: Thank you for this comment. We superscripted “-6”.

Statements: Lines 71-83: various qualitative statements are made about the different juices for different target species but no p values are provided to support those statements.

Figures 1 and 2: whereas it is always preferable to report exact p values than mere p<0.05, the conventional practice is to use one, two or three stars to indicate different degrees of statistical significance. The presence of only one star should be interpreted as in all cases p is lower than 0.05 but higher than 0.01 for instance? A three level system of indicating statistical significance, while still imperfect, would be better than the current one, using one star only.

Besides, the statistical methodology mentions ANOVA, but no information on checking ANOVA assumptions is made.

Answer: Thank you for the comments. We reworked both diagrams and marked significant differences by stars for p < 0.05. Please find attached the statistical analysis for the agar diffusion and direct contact test (supplementary materials). As you can see, most of the values show p < 0.001. By applying a system that defines the degrees of significance these values must be marked by 3 stars. This will unfortunately cause confusion to the figures. We therefore refrain from adopting a system that differentiates the degree of significance.

Further, the following assumptions were made:

  1. Each group sample is drawn from a normally distributed population (Shapiro Wilks Test)
  2. All populations have a common variance (Levine and Brown-Forsythe Tests)
  3. All samples are drawn independently of each other

We introduced additional information regarding the assumptions to the statistical section. All results were checked by the Institute of Medical Statistics, Computer Sciences and Documentation (IMSID), University Hospital Jena, Germany.

Statement: As shown by Figure 3, whereas some bacterial species were suppressed by a contact of only 60 seconds, for some a longer contact testing would have been relevant to understand whether they can be suppressed by longer contact times.

The use of only 60s for cytotoxicity assessment is much too short to allow any proper interpretation of the cytotoxicity findings. For many cytotoxicity tests hours or days are used. In this case one might speculate that to reflect the short time the juice will stay in the oral cavity a shorter time is needed, but even so, in our view it was preferable to have at least 5-10 minutes if not a few hours for a better understanding of the cytotoxicity. It is true that for such a short time (60s) the juices seem safer, but what if after 3 minutes there is no difference between them and CHX? This limitation should be discussed in the paper, as well as other limitations of the study.

Answer: We totally agree with the reviewer. Prolonged incubation times will definitely allow a more detailed point of view. But, in the present study we primarily focused upon incubation times that are comparable to those commonly reached by conventional treatments with CHX-containing mouth rinses. Under “normal” conditions the adjunct treatment of the oral cavity with CHX-containing products is of rather short duration, too. Therefore, we concluded to apply incubation times of only 60s in the present study. As already emphasized by the reviewer, prolonged incubation times will probably be associated with increased cytotoxic reactions, too. But, as shown in the present study incubation with 0.2% CHX for 60s already influenced the cell morphology to a stronger extent as observed for the berry juices. As advised by the reviewer the issue was introduced to the discussion, too. Please view lines 227 to 232.

Statement: I find the acknowledgment rather curious: the authors are thanking themselves for being authors

Answer: We agree with the reviewer and changed to a different acknowledgment.

Again, we are very thankful for all comments and suggestions and hope to have answered each question sufficiently.

Reviewer 2 Report

The manuscript by Kranz and coworkers reports the antimicrobial effect of natural berry juices on Common oral pathogenic bacteria. Specifically, the gram-positive and gram-negative bacteria (Streptococcus mutans, Streptococcus gordonii, Streptococcus sobrinus, Actinomyces naeslundii, Fusobacterium nucleatum, Aggregatibacter actinomycetemcomitans, Porphyromonas gingivalis, and Enterococcus faecalis). The mansucript is very well written, it totally makes sense and follows a nice story. It is also well referenced. I hope in the future, the exact chemicals responsible for the bioactivity be elucidated for black currant juice. This manuscript is worth of publication after the following minor corrections. Great Job!

  1. Add the control to Figure 1, So, you don’t need the stars. The bar for CHX
  2. Italize the names of all the bacteria and the plants throughout the manuscript.
  3. I believe; Antibiotics don’t charge extra for color. So, it will be great if you could add pictures of the plants (berries) in the introduction sections to aid in the visual of the paper. If possible, add the scientific names of the plants early in the manuscript other than wait until the materials and methods section. Perhaps in the new suggested pictures.

Author Response

Dear reviewer,

thank you for reviewing our manuscript and for all statements that helped us to improve the publication. We tried to answer each comment to the best of our knowledge. Please find below a detailed list of all corrections.

Statement: Add the control to Figure 1, So, you don’t need the stars. The bar for CHX.

Answer: Thank you for this comment. We defined the control by introducing the term “positive control”. Unfortunately we are not able to delete any stars because they are favored by a different reviewer.

Statement: Italize the names of all the bacteria and the plants throughout the manuscript.

Answer: We have checked all names and converted “black currant” into “blackcurrant” and “red currant” into “redcurrant”. Also we introduced the names and abbreviations of each bacterial species to the subscriptions.

Statement: I believe; Antibiotics don’t charge extra for color. So, it will be great if you could add pictures of the plants (berries) in the introduction sections to aid in the visual of the paper. If possible, add the scientific names of the plants early in the manuscript other than wait until the materials and methods section. Perhaps in the new suggested pictures.

Answer: We changed the figures to color. Also, the scientific names of each plant now appear earlier in the text. Please view lines 67-68. Unfortunately we do not have any colored pictures of the fruits, but we are planning on incorporating pictures of examined plants in upcoming publications.

Again, we are very thankful for all comments and suggestions and hope to have answered each question sufficiently.